# The Association of Redox Regulatory Drug Target Genes with Psychiatric Disorders: A Mendelian Randomization Study

**DOI:** 10.3390/antiox13040398

**Published:** 2024-03-27

**Authors:** Zhe Lu, Yang Yang, Guorui Zhao, Yuyanan Zhang, Yaoyao Sun, Yundan Liao, Zhewei Kang, Xiaoyang Feng, Junyuan Sun, Weihua Yue

**Affiliations:** 1Institute of Mental Health, Peking University Sixth Hospital, No. 51 Hua Yuan Bei Road, Beijing 100191, China; luzhe@bjmu.edu.cn (Z.L.); grzhao@bjmu.edu.cn (G.Z.); zhang_yyn@bjmu.edu.cn (Y.Z.); sunyaoyao@bjmu.edu.cn (Y.S.); yd.liao@bjmu.edu.cn (Y.L.); kangzhw@bjmu.edu.cn (Z.K.); xiaoyangfeng@bjmu.edu.cn (X.F.); sjyzhu@bjmu.edu.cn (J.S.); 2NHC Key Laboratory of Mental Health, Peking University, Beijing 100191, China; 3National Clinical Research Center for Mental Disorders, Peking University Sixth Hospital, Beijing 100191, China; 4Department of Molecular and Cellular Pharmacology, School of Pharmaceutical Sciences, Peking University, Beijing 100191, China; yang_@bjmu.edu.cn; 5PKU-IDG/McGovern Institute for Brain Research, Peking University, Beijing 100871, China; 6Chinese Institute for Brain Research, Beijing 102206, China

**Keywords:** redox regulatory drug, drug discovery, psychosis, mendelian randomization study, cognition

## Abstract

Redox regulatory drug (RRD) targets may be considered potential novel drug targets of psychosis due to the fact that the brain is highly susceptible to oxidative stress imbalance. The aim of the present study is to identify potential associations between RRD targets’ perturbation and the risk of psychoses; to achieve this, Mendelian randomization analyses were conducted. The expression quantitative trait loci (eQTL) and protein QTL data were used to derive the genetic instrumental variables. We obtained the latest summary data of genome-wide association studies on seven psychoses as outcomes, including schizophrenia (SCZ), bipolar disorder (BD), major depressive disorder (MDD), attention-deficit/hyperactivity disorder, autism, obsessive–compulsive disorder and anorexia nervosa. In total, 95 unique targets were included in the eQTL panel, and 48 targets in the pQTL one. Genetic variations in the vitamin C target (*OGFOD2*, OR = 0.784, *p* = 2.14 × 10^−7^) and melatonin target (*RORB*, OR = 1.263, *p* = 8.80 × 10^−9^) were significantly related to the risk of SCZ. Genetic variation in the vitamin E (PRKCB, OR = 0.248, *p* = 1.24 × 10^−5^) target was related to an increased risk of BD. Genetic variation in the vitamin C target (*P4HTM*: cerebellum, OR = 1.071, *p* = 4.64 × 10^−7^; cerebellar hemisphere, OR = 1.092, *p* = 1.98 × 10^−6^) was related to an increased risk of MDD. Cognitive function mediated the effects on causal associations. In conclusion, this study provides supportive evidence for a causal association between RRD targets and risk of SCZ, BD or MDD, which were partially mediated by cognition.

## 1. Introduction

With great heterogeneity in symptoms, pathogenesis, prognosis and resource allocation, psychiatric disorders are one of the primary public health issues all over the world. Psychiatric disorders involve complex molecular pathophysiological mechanisms that affect the brain’s structure, function and signaling pathways. While our understanding is continually evolving, there are some key molecular mechanisms associated with psychiatric disorders, such as neurotransmitter imbalance (including dopaminergic, serotoninergic and glutamatergic neurotransmission), genetic and epigenetic factors, neuroinflammation, neuroplasticity alterations and hormonal influences [1,2]. Previous medications for psychosis were mainly designed according to the neurotransmitter hypothesis, but new insights into the treatment or prevention of psychiatric disorders are an urgent need [3].

Oxidative stress is indispensable for normal physiological functions, yet excessive peroxide levels precipitate detrimental oxidation, fostering various pathological pathways. This imbalance results from the interplay between harmful reactive oxygen and nitrogen species (ROS/RNS) and the antioxidant defenses. Excessive ROS pose a threat to DNA, proteins and lipids [4]. Hence, antioxidants play a pivotal role in averting undue oxidative harm. The brain is a lipid-rich organ with enormous oxygen consumption and a lack of sufficient antioxidant barriers, which makes the brain highly susceptible to oxidative stress imbalance [5]. In the previous century, the connection between unbalanced oxidative stress and various neuropsychiatric diseases was found [6]. Redox homeostasis is ensured by a complex antioxidant defense system, including enzymatic (i.e., glutathione-transferase, catalase, superoxide dismutase and glutathione peroxidase) and non-enzymatic antioxidants (i.e., glutathione, kinds of vitamins, uric acid, albumin and some metal ions) [7]. Enzymatic antioxidants impede peroxide formation and scavenge excess ROS, and non-enzymatic antioxidants sequester transition metals and interrupt free-radical chain reactions [8]. In clinics, numerous abnormalities of antioxidant defense in patients with psychiatric disorders have been noted [9,10,11,12,13,14]. Additionally, previous clinical trials revealed that some redox regulatory drugs (RRDs), such as N-Acetylcysteine (NAC) and allopurinol, can improve the symptoms of psychiatric disorders (i.e., schizophrenia, depression and autism spectrum disorder), which meant that the drug targets of RRD might be possible targets for psychiatric disorders [15,16,17,18,19].

Randomized clinical trials (RCTs) are the golden criteria to explore the causal effect of numerous kinds of RRD treatment. However, these approaches are not suitable due to RCTs being expensive and uncertain. The Mendelian randomization (MR) study provides a genetic approach which applies the genetic variants associated with an exposure as the instruments to make a causal inference on the outcome. Our previous MR study demonstrated that there were bidirectional causal associations between oxidative stress injury biomarkers and psychiatric disorders, such as the associations between uric acid level and mood disorders [20].

To identify potential associations between RRD therapy and psychiatric disorders, two-sample MR analyses were conducted. These analyses integrated brain-derived molecular quantitative trait loci (mRNA expression and protein abundance quantitative trait loci [QTL]) and genome-wide association study (GWAS) findings from large-scale psychiatric genetic studies. Cognitive dysfunction is a core symptom in psychiatric disorders, and some antioxidants have a positive effect on preventing the risk of cognitive decline [21]. To uncover the underlying mechanism of the causal effect of antioxidant targets on psychiatric disorders, two-step MR analyses were performed by setting antioxidant levels and cognition function as mediators.

## 2. Materials and Methods

### 2.1. Study Design

This genetic association study used available public data from studies involving human participants, with written informed consent and approval from their respective institutional ethics review committees. We followed the Strengthening the Reporting of Observational Studies in Epidemiology Using Mendelian Randomization (STROBE-MR) reporting guidelines (Appendix A) [22]. The full period of data collection was from 7 December 2022 to 24 August 2023.

The three crucial assumptions of MR study are listed as follows: (1) there is a strong association between genetic instrumentals and target proteins, (2) there exists independence of the instruments from confounding factors and (3) instrumental variables (IVs) influence the psychiatric disorder risk only through the drug targets (Figure 1a).

### 2.2. Ethics Approval

We only used publicly available data, and hence, no ethics approval was required. The details of ethical approval and participant consent for each of the studies that contributed to the GWAS can be found in the original publications.

### 2.3. Identification of Redox Regulatory Drug Target Genes

The RRD target data were collected from 7 December 2022 to 9 January 2023. The list of RRDs commonly used for patients with psychiatric disorders was gained from our previous MR study and the review of new treatment perspectives in psychiatric disorders, which includes 18 RRDs (N-Acetylcysteine [NAC], ginkgo biloba extract, selegiline, allopurinol, polyunsaturated fatty acids [omega-3], vitamin E, vitamin C, vitamin D, vitamin A, sulforaphane, DDO-7263, curcumin, resveratrol, salvianolic acid B, metformin, coenzyme Q, melatonin and piracetam) [15,20]. The pharmacologically active protein targets and corresponding encoding genes of these RRDs were retrieved from the DrugBank (https://go.drugbank.com/, accessed from 7 December 2022 to 9 January 2023) and ChEMBL databases (https://www.ebi.ac.uk/chembl/, accessed from 7 December 2022 to 9 January 2023) separately (Figure 1b, Appendix A).

### 2.4. Genetic Instruments for Redox Regulatory Drug Target Genes

Two QTL databases were utilized to derive genetic IVs. The first dataset is the expression QTL (eQTL) data from The Genotype-Tissue Expression project (GTEx, version 8, data from 139 to 255 subjects on genotypes; its samples mostly stemmed from European-ancestry individuals [about 85%]); the second dataset is the Religious Orders Study and Rush Memory and Aging Project (ROSMAP, data from 376 subjects from the dorsolateral prefrontal cortex [DLPFC] of postmortem brain samples) which provides the protein QTL (pQTL) summary data. This study focused on the psychiatric disorders, so we extracted eQTL data from 13 brain tissues of the GTEx results. We only included cis single nucleotide polymorphisms (cis-SNPs) associated with gene expression. Genetic variants associated with the expression of actionable drug targets (eQTL, *p* < 1 × 10^−4^) were selected for subsequent analysis [23]. Because one gene corresponds to one instrument, and the selected instrument shows the most significant association with the gene in the GTEx eQTL summary statistics, LD clumping is not applicable for this dataset. For the pQTL information, ROSMAP performed a pQTL analysis in the prefrontal cortex to identify genetic variants associated with protein abundance in the human brain. A total of 912,253 SNP–protein expression pairs were included in the ROSMAP pQTL dataset, and SNPs were extracted if they showed significant associations with the expression of target proteins (pQTL *p* < 0.05). We performed LD clumping by using the *clump_data()* function in the *TwoSampleMR* R package (version 0.5.6), with the use of default LD clumping parameters, i.e., the clumping r2 cutoff was set to 0.001 and “EUR” was selected as the LD reference panel (Figure 1b, Appendix A).

A larger *F* statistic indicated a stronger instrument strength; *F* statistics were used to test for weak IVs. The *F* statistics of all SNPs included in the two-sample MR analysis were evaluated, and the formula is as follows: F*_j_* = γ^*_j_*^2^/*σX_j_*^2^ (γ^*_j_* = SNP–exposure association; *δ_Xj_* = standard error of the SNP–exposure association) [24]. The IV with F statistics less than 10 was excluded (Table 1).

### 2.5. Outcome Data

We collected the most recent case-control GWAS summary data for seven psychiatric disorders. These disorders include schizophrenia (SCZ) [25], bipolar disorder (BD) [26], major depressive disorder (MDD) [27], attention-deficit/hyperactivity disorder (ADHD) [28], autism spectrum disorder (ASD) [29], obsessive–compulsive disorder (OCD) [30] and anorexia nervosa (AN) [31]. All of the individuals were of European descent. The GWAS summary statistics were downloaded from the Psychiatric Genomics Consortium (PGC) website (https://pgc.unc.edu/, accessed from 9 January 2023 to 24 August 2023) (Figure 1b; the detailed information of these GWASs is described in Appendix A).

### 2.6. Mendelian Randomization

All analyses were performed by using R software (version 4.1.3, R Foundation for Statistical Computing, Vienna, Austria), and the *TwoSampleMR* R package (version 0.5.6) was used to perform two-sample MR analysis. The two-sample MR framework requires two datasets to conduct MR analysis. In this study, the cis-eQTL and pQTL data were used as genetic proposed instruments (exposure), and the GWASs were used as the outcome trait data. MR tests the relationship between gene expression and diseases (or traits) by using genetic variants associated with gene expression (exposure) as IVs and GWASs as outcomes. MR could investigate if a change in gene expression has causal effects on diseases or traits. For proposed instruments with one SNP, the Wald ratio was used. For proposed instruments containing more than one SNP, inverse-variance-weighted (IVW) MR was conducted (Figure 1).

In total, 95 unique targets were included in the eQTL panel, and the multiple correction level for the MR analysis result was set at *p* < 5.78 × 10^−6^ (*p* = 5.78 × 10^−6^ (0.05/(95 × 13 × 7))). For the pQTL panel, 48 unique targets were included, and the correction threshold was set at *p* < 1.49 × 10^−4^ (*p* = 1.49 × 10^−4^ (0.05/(48 × 7))). No further correction was applied for the pQTL MR analysis as only one pQTL panel was included in this study. The odds ratios (ORs), beta and 95% confidence intervals (CIs) were used to present the causal effect (Figure 1b).

### 2.7. Sensitivity Analyses

The single-cell RNA sequencing (RNA-seq) data were used to detect the gene expression in specific brain cell clusters, which was gained from the gene expression in single cells of the postmortem tissues from 16 ASD patients and 16 healthy controls (https://cells.ucsc.edu/?ds=autism, accessed on 15 March 2024). A total of 11 types of neurons and six types of glial cells were included [32].

For the significant MR results, we performed Bayesian colocalization analysis using the *coloc* (version 5.2.1) R package for validation. Colocalization analysis can be used in a complementary manner to MR in that it can reduce linkage disequilibrium bias and confirm whether a causal effect is driven by the same variant. This approach operates under the following assumptions: (1) Each trait has at most one causal genetic instrument. (2) The likelihood of a genetic instrument being causal is independent of other genetic instruments in the analysis. (3) The colocalization analysis includes all causal genetic variants, whether genotyped or imputed. Based on these assumptions, there are five hypotheses for each analysis. H4 suggests the presence of a shared causal genetic instrument for both traits (Table 1). We used default prior probability: p1 = 1 × 10^−4^; p2 = 1 × 10^−4^; p12 = 1 × 10^−5^. p1, p2 and p12 represent the prior probabilities of a significant association between an SNP in the tested region and the expression of the tested gene, the tested outcome or both, respectively. Variants ± 100 kb of the top SNP were included. Default priors were used for analysis. The LD reference panel utilized was the European ancestry dataset from the 1000 Genomes v3. We evaluated the evidence of colocalization by posterior probability for hypothesis 4 (PP.H4), which indicates the presence of associations for both the drug target and the disease, driven by the same causal variant(s). Associations with a PP.H4 greater than 0.5 were considered indicative of likely colocalization, as this assigns the highest probability to hypothesis 4 being accurate. The *LocusCompare* R package (version 1.0.0) was used to visualize the colocalization results [33] (Figure 1).

### 2.8. Two-Step Mendelian Randomization for Mediation Analysis

In employing the two-step MR, we aimed to evaluate the effect of potential mediators on the relationship between exposure and outcome. This analysis discerns both direct and indirect effects within the total effect. The direct effect delineates the exposure’s influence on the outcome, independent of intermediary variables, while the indirect effect denotes the exposure’s impact on the outcome through intermediary variables. This statistical approach enables the exploration of intricate relationships among variables, elucidating how intermediary factors influence exposure–outcome associations. For the significant MR results, the two-step MR analysis was conducted to assess whether the antioxidant level in vivo or the cognitive function has a role in the mediating pathway between the targets and psychiatric disorders. The GWASs’ summary statistics of antioxidants were obtained from the open database that has been made publicly available (IEU OPEN GWAS PROJECT: https://gwas.mrcieu.ac.uk/, accessed from 9 January 2023 to 24 August 2023). Five GWASs’ summary statistics of the cognition domain (fluid intelligence: fluid intelligence score; trial marking: duration to complete numeric path and duration to complete alphanumeric path; pairs matching: number of incorrect matches in round and number of correct matches in round; numeric memory: maximum digits remembered correctly; prospective memory: prospective memory result) were gained from the UK biobank. In order to avoid the bias of population heterogeneity, only the European population summarized data were adopted. In the first step, genetic instruments for RRD targets were used to estimate the causal effect of the exposure on mediators. In the second step, genetic instruments for the mediators were used to assess the causal effect of mediators on psychiatric disorder risk. Both steps in the MR results were significant (*p* < 0.05) and overlapping was regarded as a mediator (Figure 1).

## 3. Results

### 3.1. Genetic Instrument Selection

A total of 110 genes whose encoded protein activity has been experimentally shown to be modified by one or more RRDs were identified from the two databases; four RRDs (Salvianolic acid B, Piracetam, DDO-7263 and Sulforaphane) were excluded because there were no available data. Next, 15 genes in the eQTL panel and 62 genes in the pQTL panel were filtered out because they were absent in the summary statistics. Finally, 95 unique targets were included in the eQTL panel, and 48 unique targets were included in the pQTL panel (Figure 1b).

To avoid potential confounding, we investigated each IV in the PhenoScanner (http://www.phenoscanner.medschl.cam.ac.uk/, accessed from 9 January 2023 to 24 August 2023) GWAS database to assess any previous associations (*p* < 5 × 10^−8^) with the seven psychiatric disorders.

### 3.2. Causal Effects of Redox Regulatory Drug Targets on Psychiatric Disorders

According to a significance threshold of *p* < 5.78 × 10^−6^ for the eQTL level and *p* < 1.49 × 10^−4^ for the pQTL level, genetic variations in the vitamin C target (cofactor, *OGFOD2* in cerebellum, OR = 0.784, 95% CI = 0.715–0.859, *p* = 2.14 × 10^−7^) and the melatonin target (agonist, *RORB* in cerebellum, 1.263, 95% CI = 1.167–1.368, *p* = 8.80 × 10^−9^; additionally, there were many nominally significant results which could confirm the above results: hypothalamus, *p* = 1.04 × 10^−2^; putamen basal ganglia, *p* = 4.40 × 10^−3^) were significantly related to the risk of SCZ. For bipolar disorder, genetic variation in the vitamin E (PRKCB in ROSMAP, OR = 0.248, 95% CI = 0.152–0.403, *p* = 1.24 × 10^−5^; rs4967960 was excluded due to the *F* statistic being less than 10) target was related to an increased risk of BD. Moreover, variations in the vitamin C target (cofactor, *P4HTM*) showed a link to an increased risk of MDD. Specifically, in the cerebellum and cerebellar hemisphere, the OR was 1.071 (95% CI = 1.043–1.100, *p* = 4.64 × 10^−7^) and 1.092 (95% CI = 1.053–1.132, *p* = 1.98 × 10^−6^), respectively. Additionally, there were other regions with nominally significant results supporting this finding: cortex (*p* = 4.70 × 10^−3^), frontal cortex (*p* = 4.80 × 10^−2^), hippocampus (*p* = 1.08 × 10^−2^), hypothalamus (*p* = 4.80 × 10^−2^), putamen basal ganglia (*p* = 1.08 × 10^−2^) and ROSMAP (*p* = 8.42 × 10^−3^). All of these results pointed toward an increased risk of MDD. According to the PhenoScanner website, one IV was excluded (rs28768122 located in *OGFOD2* in the frontal cortex tissue) (Figure 2a, Appendix A).

In addition to the significant results, there were lots of suggestive results (with a *p* value less than 0.05) which were reasonable due to the consistency across different brain regions. For instance, genetic variations in the metformin target, *PRKAB1* (inducer activator), are linked to the risk of SCZ. This association was observed at both the eQTL level (caudate basal ganglia: *p* = 6.28 × 10^−4^; cerebellum: *p* = 5.46 × 10^−4^; putamen basal ganglia: *p* = 5.89 × 10^−4^) and the pQTL level (*p* = 2.11 × 10^−3^). Similarly, variations in the ginkgo biloba extract target, *NOS2* (inhibitor), are associated with the risk of BD and MDD across various brain regions. Genetic variations in the vitamin C target, *P4HTM* (cofactor), are linked to the risk of ADHD across different brain regions. Genetic variations in the metformin target (*ETFDH*, inhibitor), the ubiquinone target (*SDHA*, cofactor) and the vitamin C target (*EGLN1*, cofactor) are associated with the risk of ADHD across multiple brain regions. Genetic variations in the ubiquinone target (*NDUFV3*, cofactor) are related to the risk of AN, and variations in the metformin target (*PRKAB1*, inducer activator) are associated with the risk of OCD (Appendix A).

### 3.3. Gene Expression Level in Single-Cell Clusters

The single-cell RNA-seq data revealed that *OGFOD2* was not expressed in the particular cell cluster. This absence of expression was likely because the single-cell RNA-seq data were obtained from tissues of the prefrontal cortex and anterior cingulate cortex, whereas *OGFOD2* was found to be highly expressed in the cerebellum. (See Figure 2b, GTEx Portal.) The RORB gene was highly expressed in the layer 4 excitatory neurons, fibrous astrocytes and protoplasmic astrocytes (Figure 2c). The *PRKCB* and *P4HTM* gene was highly expressed in the NRGN-expressing neurons (Figure 2d,e).

### 3.4. Colocalization Results

To ensure the reliability of the two-sample MR results, we conducted colocalization analyses. These analyses focused on examining the relationship between the expression of potential targets in specific tissues and psychiatric disorders. For schizophrenia, we looked at the expression of *OGFOD2* and *RORB* in the cerebellum. For bipolar disorder, we examined the protein encoded by *PRKCB*. For major depressive disorder, we investigated *P4HTM* expression in both the cerebellum and cerebellar hemisphere.

We found strong evidence suggesting shared genetic variants between certain gene expressions in specific brain regions and psychiatric disorders. There was a high posterior probability (92.0%) that a genetic variant (rs28569885) was shared between *OGFOD2* gene expression in the cerebellum and SCZ. Similarly, there was a 98.1% posterior probability of a shared genetic variant (rs514209) between *RORB* gene expression in the cerebellum and SCZ. We also observed a 97.0% posterior probability of a shared genetic variant (rs120908) between PRKCB protein expression and BD. Furthermore, there was a 58.6% posterior probability of a shared genetic variant (rs6769821) between *P4HTM* gene expression in the cerebellum and MDD, and a 52.0% posterior probability of a shared genetic variant (rs4384984) between *P4HTM* gene expression in the cerebellar hemisphere and MDD (Figure 3, Table 1).

### 3.5. Two-Step MR

Aimed to reveal the potential mechanism of the RRD target for psychiatric disorders, two-step MR studies were conducted. We firstly hypothesized that the causal effects of RRD targets on psychiatric disorders were mediated by the antioxidant itself. In the first step, we used genetic instruments for important targets in specific tissues to investigate how gene or protein expression affects antioxidants like vitamin C and vitamin E (though we did not have access to GWAS data on melatonin). Then, in the second step, we examined how these antioxidants influence psychiatric disorders (including SCZ, BD and MDD). The antioxidant with significant *p* values (*p* < 0.05) in both of the steps was regarded as the mediator. However, the antioxidant levels themselves did not have mediated effects on the causal association between redox regulatory drug targets and psychiatric disorders (Figure 4a, Appendix A).

Accounting for the known cognitive influence effect of RRDs, we then hypothesized that the causal effects of RRD targets on psychiatric disorders were mediated by their cognitive influence effect. Following the same process, causal effects of gene or protein expression on cognition and the causal effects of cognition on psychiatric disorders were evaluated. We found that cognitive function exerted mediating or suppressing effects. The fluid intelligence score and pairs matching have a suppressing effect on the direct link between *OGFOD2* and SCZ, with a proportion of 11.13% and 8.59%, respectively. Additionally, the mediating effect of trial marking influences the direct link from *RORB* to SCZ, with numeric and alphanumeric paths at 7.58% and 12.04%, respectively. Furthermore, fluid intelligence, pairs matching and trial marking all have a suppressing effect on the direct link from *P4HTM* to MDD, with varying percentages depending on the brain region (Figure 1c and Figure 4b, Appendix A).

## 4. Discussion

In this study, we used the QTL and GWAS summary data in the two-sample MR analyses to infer potential causal effects of RRD targets on psychiatric disorders. The findings indicated that higher levels of *OGFOD2* gene expression (target of vitamin C, cofactor) and lower levels of *RORB* gene expression in the cerebellum (target of melatonin, agonist) are linked to a decreased risk of SCZ. Similarly, higher levels of PRKCB protein in the DLPFC (target of vitamin E) are associated with a lower risk of BD. Additionally, lower levels of *P4HTM* gene expression in the cerebellum (target of vitamin C, cofactor) are associated with a decreased risk of MDD. Next, the two-step MR analyses indicated that these causal effects of targets on the risk of psychiatric disorders were partly mediated by cognitive function, as well as the antioxidants themselves. The colocalization analyses also provided a certain level of support for the significant MR results.

*OGFOD2*, 2-oxoglutarate and iron-dependent oxygenase domain-containing 2, is related to L-ascorbic acid binding activity, dioxygenase activity and iron ion binding activity. It plays a role in oxidoreductase activity, specifically acting on paired donors, with the incorporation or reduction of molecular oxygen. *OGFOD2* has been proven to be associated with a risk of SCZ [34]. The protein encoded by the *RORB* (retinoid-related orphan receptor beta) gene has the ability to help regulate the expression of some genes involved in circadian rhythm. RORbeta knockout mice displayed reduced anxiety and learned helplessness-related behaviors [35], which confirmed the above result that lower expression of the *RORB* gene is associated with a decreased risk of SCZ. Moreover, Mansour HA et al. evaluated the associations between circadian gene polymorphisms and psychosis, and found that *RORB* gene polymorphism is associated with both SCZ and BD [36].

PRKCB (protein kinase C beta) is a family of serine- and threonine-specific protein kinases that can be activated by calcium and second messenger diacylglycerol. PRKCB has been reported to be involved in the regulation of the B-cell receptor signalosome, oxidative stress-induced apoptosis, androgen receptor-dependent transcription regulation, insulin signaling and endothelial cell proliferation. PRKCB insufficiency is related to immune cell-mediated autoimmune disorders [37,38]. Compared with healthy controls, the protein and mRNA expression of protein kinase C in the postmortem brain (including PKCβI and PKCβII) of bipolar subjects was significantly decreased [39]. Our findings suggest that a higher level of PRKCB protein is associated with a decreased risk of BD, which is in consistent with cadaveric brain research.

The product of the *P4HTM* (Prolyl 4-Hydroxylase, Transmembrane) gene belongs to the family of prolyl 4-hydroxylases. P4HTM is the crucial enzyme that modulates the hypoxia inducible factor response and helps to maintain cellular oxygen homeostasis. Experimental evidence suggests that *P4HTM* plays a role in social behaviors and anxiety. The *P4htm* knockout mice were more active than the wild-type mice. However, the *P4htm* knockout mice showed normal spatial learning and memory in the Morris swim task. These knockout mice also exhibited increased interaction with fellow mice in social behavior tests and lacked the typical behavioral despair response in the forced swim and tail suspension test [40]. In this study, a lower expression of the *P4HTM* gene was associated with a decreased risk of MDD, which is in consistent with the experimental evidence. 

Expression analysis of single-cell sequencing data provided further evidence of the potential roles of target genes in the brain. Excitatory neurons have been identified as playing a critical role in the pathogenesis [41]. Additionally, astrocytes, the most abundant glial cells in the brain, have also been found to be associated with schizophrenia [42]. Neurogranin, encoded by the *NRNG* gene, is a calmodulin-binding protein that is highly enriched in the postsynaptic compartments of neurons. It is believed that the hypofunction of neurogranin may lead to experience-dependent developmental deficits in excitatory synapse connectivity or cognitive impairment, thereby potentially contributing to the pathogenesis of psychosis [43].

According to our findings, the suggested RRDs are vitamin C (ascorbic acid), vitamin E and melatonin. Ascorbic acid, as a neuroprotective compound, is highly concentrated in the brain and regulates the function of neurons and synapses [44]. It has been reported to enhance dopamine-mediated behavioral effects [45]. Moreover, the pretreatment of haloperidol or sulpiride prevented the antidepressant-like effect elicited by ascorbic acid in the tail suspension test [46]. Ascorbic acid is also released in the striatum and nucleus accumbens after methamphetamine administration [47,48]. The findings of these animal studies were in consistent with our MR results. Vitamin E is anchored in a lipid membrane. The hydroxyl group on its six-membered ring is located outside of the membrane towards the water environment, where it scavenges ROS [49]. The neuroprotective role of vitamin E in the brain, therefore, has been linked to neurogenesis, neuronal differentiation, hippocampal synaptic function and cell signaling pathways [50]. Additionally, the antioxidant capacity of vitamin E has been reported to be linked to vitamin C [51]. Melatonin is known to modulate circadian rhythms and is involved in sleep regulation, reproductive physiology and immunological regulation [52].

Therefore, antioxidant levels in vivo may mediate the direct effect of RRD target gene expression on psychosis. Unfortunately, the two-step MR analyses did not find a significant mediation effect of the antioxidant levels. Oxidative stress is also a crucial risk factor for the pathogenesis and progression of impaired cognitive function [53]. Hence, we hypothesized that cognitive function exerted a mediation effect. In our two-step MR analyses, we found that higher levels of *OGFOD2* gene expression and lower levels of *RORB* gene expression in the cerebellum were linked to a decreased risk of cognitive dysfunction. Similarly, lower levels of *P4HTM* gene expression in the cerebellum were also associated with a decreased risk of cognitive dysfunction. These findings align with the associations observed between gene expression and the risk of three psychotic disorders. The two-step MR results provide evidence that the causal effect of target gene expression on psychotic disorders is partly mediated by cognition. Although we observed a positive association between RRD target gene expression and psychoses, the antioxidants as medication for psychiatric disorders lack high-level evidence due to the low quality and sample size [16,49,54,55].

The main limitation of this study is the assumption that changes in protein-coding gene expression may reflect changes in protein levels and/or activity in the eQTL panel, which may not always be the case. The non-coding RNAs are not considered. Additionally, the single-cell RNA-seq data were obtained from the prefrontal cortex and anterior cingulate cortex tissues, while the results in this study were mainly found in the cerebellum tissue; therefore, the evidence was insufficient. Moreover, even though the expression level of RRD targets might be in fact indicative of the relationship between oxidative stress and the disease that was examined, there are many ways in which RRDs may act, besides the interaction with specific genes. Finally, it is difficult to extrapolate the actual effect of drug exposure from genetic analyses, as the effect sizes estimated from the MR analysis may not reflect the true effect of drug exposure.

## 5. Conclusions

In conclusion, this study provides supportive evidence for the causal association between redox regulatory drug targets and the risk of SCZ, BD or MDD, which were partially mediated by cognition. Future studies should be conducted to clarify the underlying mechanistic pathways between redox regulatory drug targets and psychiatric disorders. The present study also exemplifies how the MR design may be a promising tool for finding new indications for approved drugs. This method allows us to test interventions that are otherwise costly, time-consuming or, in other ways, impractical, and should be considered a screening instrument in the drug development phase.

## Figures and Tables

**Figure 1 antioxidants-13-00398-f001:**
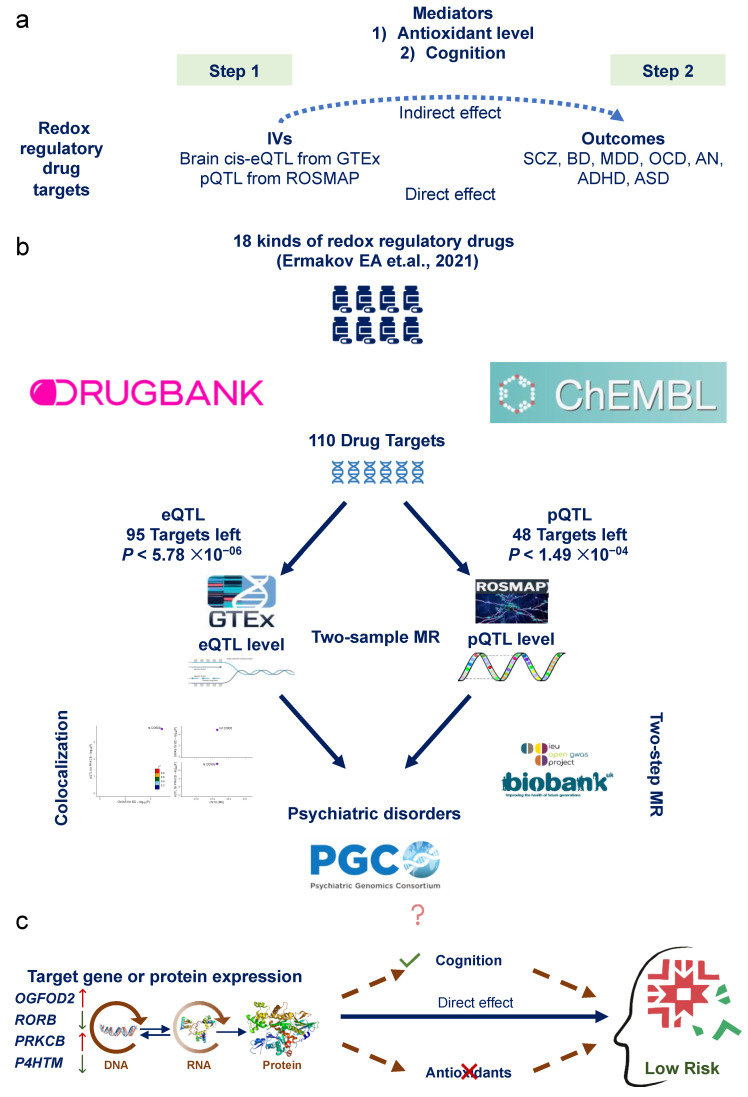
Study design and flow chart. (**a**) Study design; (**b**) flow chart [15]; (**c**) potential pathway. IVs, instrumental variables; MR, Mendelian randomization; ROSMAP, Religious Orders Study and Rush Memory and Aging Project; PGC, Psychiatric Genomics Consortium; SCZ, schizophrenia; BD, bipolar disorder; MDD, major depressive disorder; OCD, obsessive–compulsive disorder; AN, anorexia nervosa; ADHD, attention-deficit/hyperactivity disorder; ASD, autism spectrum disorder; eQTL, expression quantitative trait loci; pQTL, protein quantitative trait loci.

**Figure 2 antioxidants-13-00398-f002:**
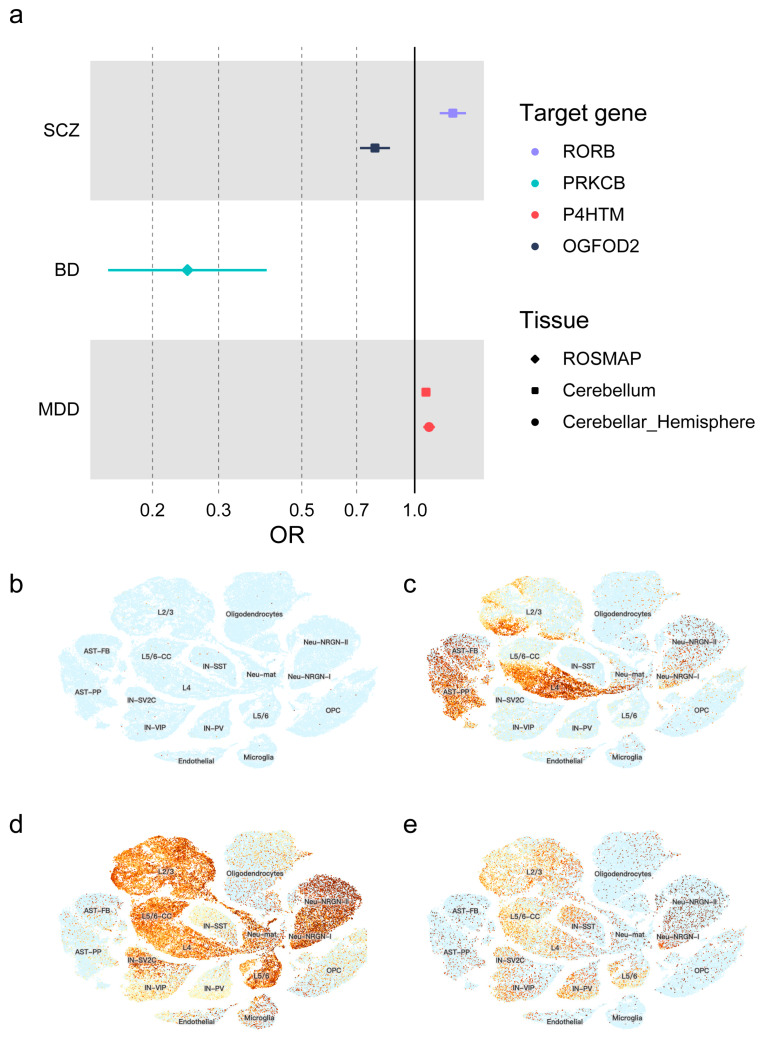
Significant results of two-sample Mendelian analyses. (**a**) Forest plot of the significant results. SCZ, schizophrenia; BD, bipolar disorder; MDD, major depressive disorder; OR, odds ratio. (**b**–**e**) Gene expression patterns from single-cell RNA-seq data of significant targets. (**b**) *OGFOD2*; (**c**) *RORB*; (**d**) *PRKCB*; (**e**) *P4HTM*. Cell type abbreviations: L2/3, upper-layer excitatory neurons; AST-FB, fibrous astrocytes; AST-PP, protoplasmic astrocytes; L5/6-CC, deep-layer cortico-cortical excitatory projection neurons; IN-SV2C, SV2C-expressing interneurons; IN-VIP, VIP interneurons; Oligodendrocytes; IN-SST, Somatostatin interneurons; L4, layer 4 excitatory neurons; Neu-mat, immature neurons; L5/6, deep-layer cortico-subcortical excitatory projection neurons; IN-PV, parvalbumin interneurons; Endothelial, endothelial neurons; Microglia; Neu-NRGN-II, NRGN-expressing neurons; Neu-NRGN-I, NRGN-expressing neurons; OPC, oligodendrocyte precursor cells.

**Figure 3 antioxidants-13-00398-f003:**
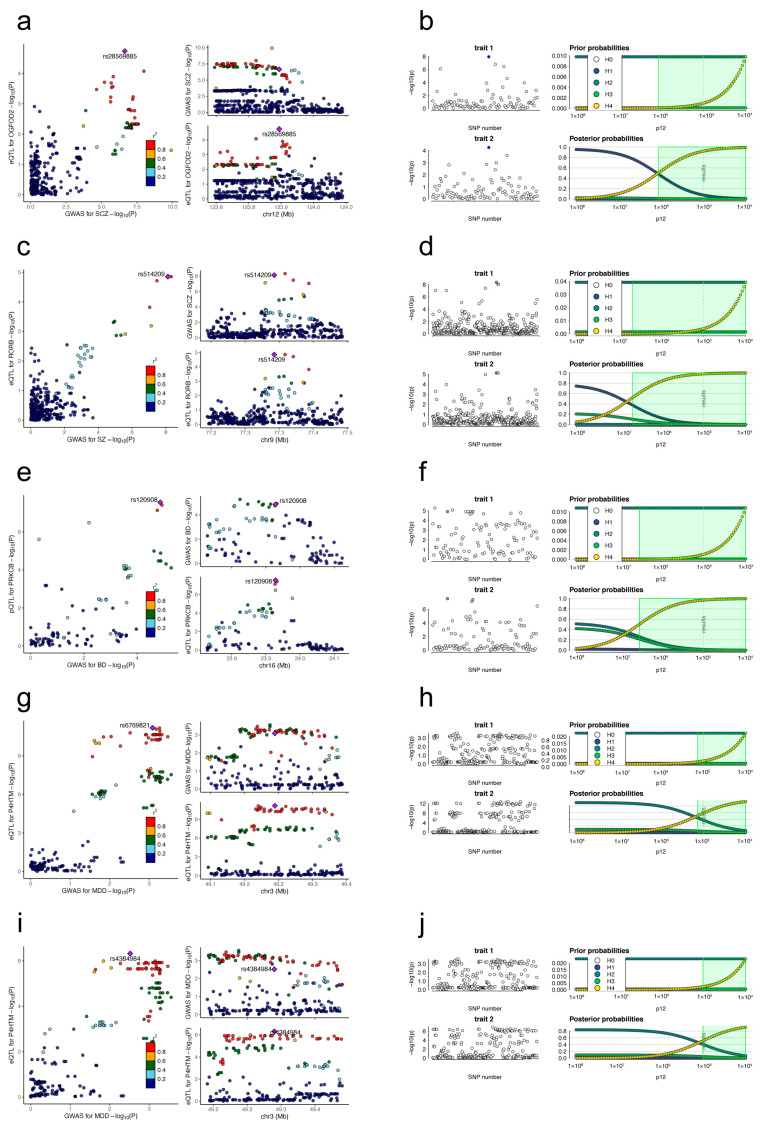
The results of the colocalization analyses. (**a**,**c**,**e**,**g**,**i**) LocusCompare plots. The purple diamond represents the target SNP (IV) in the colocalization analysis. All other SNPs are color-coded according to the strength of linkage disequilibrium (as measured by r^2^) with this index SNP. (**b**,**d**,**f**,**h**,**j**) Sensitivity plot of the colocalization analysis. The p12 indicates the prior probability of the hypothesis. The green zone represents the set of p12 values for which the provided rule H4 > 0.5 applies. Trait 1 represents the psychotic disorder, and trait 2 represents the target. SCZ, schizophrenia; BD, bipolar disorder; MDD, major depressive disorder; IV, instrumental variable; SNP, single nucleotide polymorphism; eQTL, expression quantitative trait loci; pQTL, protein quantitative trait loci.

**Figure 4 antioxidants-13-00398-f004:**
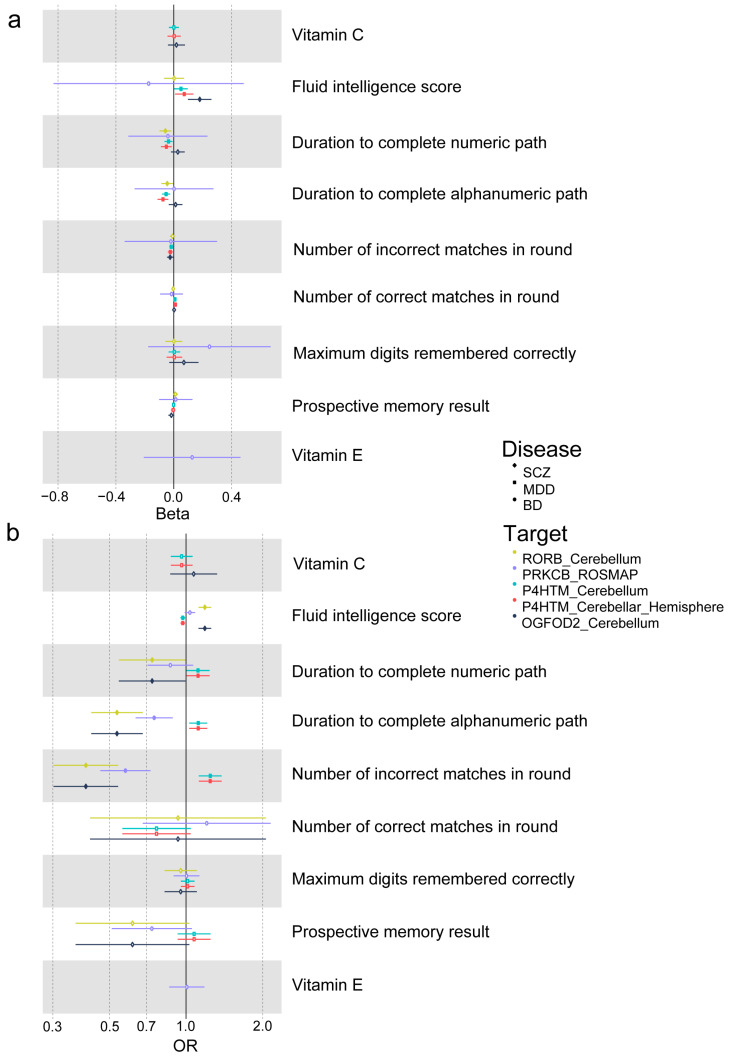
Summary results of two-step Mendelian analyses. (**a**) Step 1: Mendelian analyses for the causal effects of redox regulatory drug target genes or protein expressions on mediators; (**b**) step 2: Mendelian analyses for the causal effects of mediators on SCZ (rhomboid), BD (circle) and MDD (square). Solid graphs mean the results are significant. *OGFOD2* gene expression in the cerebellum is in blue, *RORB* gene expression in the cerebellum is in yellow, PRKCB protein expression is in purple, *P4HTM* gene expression in the cerebellum is in green and *P4HTM* gene expression in the cerebellar hemisphere is in red. SCZ, schizophrenia; BD, bipolar disorder; MDD, major depressive disorder; OR, odds ratio.

**Table 1 antioxidants-13-00398-t001:** Summary of Bayesian colocalization for redox regulatory drug targets and psychiatric disorders.

Targets	Tissues	Psychiatry Disorders	IVs	F	H0	H1	H2	H3	H4
*OGFOD2*	Cerebellum	Schizophrenia	rs28569885	19.45	1.11 × 10^−3^	7.69 × 10^−2^	3.83 × 10^−5^	1.74 × 10^−3^	9.20 × 10^−1^
*RORB*	Cerebellum	Schizophrenia	rs514209	20.03	5.20 × 10^−5^	1.52 × 10^−2^	1.76 × 10^−5^	4.16 × 10^−3^	9.81 × 10^−1^
PRKCB	DLPFC	Bipolar disorder	rs120908	54.54	6.82 × 10^−4^	6.02 × 10^−3^	1.57 × 10^−2^	1.29 × 10^−2^	9.70 × 10^−1^
*P4HTM*	Cerebellum	Major depressive disorder	rs6769821	28.01	3.11 × 10^−6^	3.37 × 10^−7^	3.74 × 10^−1^	4.00 × 10^−2^	5.86 × 10^−1^
*P4HTM*	Cerebellar Hemisphere	Major depressive disorder	rs4384984	30.85	2.91 × 10^−2^	3.28 × 10^−3^	4.03 × 10^−1^	4.49 × 10^−2^	5.20 × 10^−1^

The columns H0–H4 display the posterior probabilities of different scenarios: H0 = neither the target nor the psychiatric disorder has a genetic association with the IV, H1 = only the psychiatric disorder has a genetic association with the IV, H2 = only the target has a genetic association with the IV, H3 = both the target and the psychiatric disorder are associated but with different causal variants, and H4 = both the target and the psychiatric disorder are associated and share a single causal variant. Genes are displayed in italics. DLPFC, dorsolateral prefrontal cortex; IVs, instrumental variables.

## Data Availability

This study was based on publicly available summarized data on the GTEx (https://www.gtexportal.org/home/) and PGC (https://www.med.unc.edu/pgc/) websites.

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
