# Peer review of "The Association of Redox Regulatory Drug Target Genes with Psychiatric Disorders: A Mendelian Randomization Study"

_antioxidants, 2024, doi:10.3390/antiox13040398_

Round 1
Reviewer 1 Report
As highlighted in the review, language mistakes and the use of unfamiliar representations of results make the review very difficult to understand. I would appreciate that the Authors provide a guidance to the reader in interpreting the figures. In particular, Figure 3 is very hard to follow.
This manuscript makes use of GWAS data obtained in single cell analysis to investigate the relationship between a large array of psychiatric diseases and oxidative stress. The key of the study is the analysis of gene and protein expression of a number of targets of redox regulatory genes by using Mendelian randomization studies.
The methods applied are complex, but they certainly partially reach the aim of the study without the need for running new experiments. The main limitation of the study is in my opinion the assumption that gene expression that matters is that of protein-coding genes, whereas non-coding RNAs are not even considered. Moreover, even though the expression level of redox-regulating drug targets might be in fact indicative of the relationship of oxidative stress with the disease that was examined, there are many ways by which redox-regulating drugs may act, besides the interaction with specific genes.
The methods applied in this study are sophisticated and the way of representing results may be not very familiar to the readers. It would be very useful to guide the readers in the interpretation of figures.
There are dozens of English grammatical mistakes, which, together with the complexity of the way results are visualized and the abundancy of acronyms, can make very difficult to understand the meaning of results and of discussion. For instance: “The indicative of likely colocalizations with a posterior probability 92.0% of a shared genetic variant…” “Follow the same process, causal effects of gene or protein expression on cognition…”
Reviewer 2 Report
This is a research article with adequate novelty. However, several ponits should be addressed.
The Abstract is quite complex, including several syntax/grammar errors. Accordingly, the authors used a complex language in the Introduction section with many syntax/grammar errors. Also, certain methodological issues should be revised. The Figures legends are too long, while their resolution should be improved. A more simple language should be used in the results section in order to be more easily understood by the readers. Again, the discussion is quite compex with too lon sentences.
This is a research article with adequate novelty and quite high quality. However, several ponits should be addressed.
- The abstract should be re-organized by including distinct separate sections (e.g. background, methods, results, conclusions).
- The are several syntax/grammar errors in the Abstract. For example, the first sentence is quite confusing and it could be re-written as "The redox regulatory drug (RRDs) targets may be considered as potential novel drug targets of psychoses disorders due to the fact that brain is highly susceptible to oxidative stress imbalance.".
- The 2nd sentence of the Abstract is also confusing and it could start as "The aim of the present study is ....".
- In the Abstract, GWAS should be explained.
- The Introduction section is a bit small. The authors should include a brief description/paragraph of the basic molecular pathophysiological mechanisms taking place in psychiatric disorders by adding also relevant references.
- The authors used a complex language in the Introduction section with many syntax/grammar errors (for example lines 66-70). Some sentences are quite long and they should be split into 2 sentences such as the previous one (lines 66-70).
- In Materials and Methods section, an ethical approval code is missing. Moreover, the legend of Figure 1 is too long.
- In Materials and Methods section, it is not clear what statistical analysis softwares were used.
- A more simple language should be used in the Materials and Methods section. Some sentences are too long and they should be split into 2 sentences (e.g. lines 119-122, 147-153).
- Again, a more simple language should be used in the Results section. Some sentences are too long and they should be split into 2 sentences (e.g. lines 227-232, 255-267, 269-272, 277-281, 294-301, 307-311, 320-329,
- The legend of Figure 2 is too long and it should be condensed a bit. Moreover its resolution is very poor, especially for b, c, d and e figure parts.
- - The legend of Figure 3 is too long and it should be condensed a bit. Moreover its resolution is very poor, and it should be improved.
- The resolution of Figure 4 is very poor and the incited words should be increased.
- - The are several syntax/grammar errors in the Discussion section with long sentences which may confuse the readers. For example, some sentences should be split into two sentences (e.g. lines 342-347, 352-356, 375-380, 392-396).
- The authors should try to add some another recent studies on their topic from the last 2-3 years.
Reviewer 3 Report
I have one major comments. In line 43 citation is necessary. The sentence regarding the current therapy is not correct. However the other thing that the antioxidants drug or RRDs has in the literature over 20 years without any miracle. Some of this is can not be apply in CNS therapy.
line 43 add a reference and modify the sentence
Reviewer 4 Report
The present study explores potential connections between redox regulatory drug therapy and psychiatric disorders. The authors used two-sample Mendelian randomization analyses, integrating public data from brain-derived mRNA expression and protein abundance quantitative trait loci and genome-wide association studies from extensive psychiatric genetic studies. The study suggests an association among RRD-targets and risk of schizophrenia, bipolar disorder or major depressive disorder, related factors mediated by the cognition. The manuscript is well-written and provides important evidence. Some minor comments could be taken into account for improvement that I would like to be addressed.
The authors could stress more the Mendelian randomization study, the impact of this process, the difference between one-step and two steps.
The suggested drugs targeting redox pathways are on line with the existed which implicate in cellular processes or neurotransmitter systems and inflammation. How they could influence the overall neurobiology of psychiatric conditions?
It would be interesting for the authors to estimate a few different parameters and plot the molecular pathway and proposed theoretical attenuation of symptoms given therapeutic intervention
The authors could increase the quality of Figure 2b-e and Figure 3.
The authors could increase the quality of Figure 2b-e and Figure 3.
Reviewer 5 Report
The association of redox regulatory drugs with psychiatric disorders was important in this research field.
The manuscript was clearly and simply written. There were some minor concerns.
The first paragraph in Introduction, were not any references needed?
Line 103 and 109, correct "redox regulatory drugs" to "RRD".
Round 2
Reviewer 1 Report
I appreciated the effort to ameliorate the clarity of the manuscript
No further comments needed
Reviewer 2 Report
The authors have significantly improved their manuscript. I would like to thank you for their trust to my suggestions/comments.
The authors have significantly improved their manuscript. I would like to thank you for their trust to my suggestions/comments.
Reviewer 3 Report
The authors have made corrections to the manuscript based on the reviewers' comments. Therefore, the manuscript is now acceptable for publication in its present form.
The authors have made corrections to the manuscript based on the reviewers' comments. Therefore, the manuscript is now acceptable for publication in its present form.
Reviewer 4 Report
No major comments in this second round. The authors have addressed my comments and suggestions in the revised manuscript.
The authors have addressed my comments and suggestions in the revised manuscript.
Reviewer 5 Report
The authors improved the manuscript according to the reviewer's comments.
The manuscript was sufficiently improved.